# On Time: A Qualitative Study of Swedish Students’, Parents’ and Teachers’ Views on School Attendance, with a Focus on Tardiness

**DOI:** 10.3390/ijerph17041430

**Published:** 2020-02-23

**Authors:** Maria Warne, Åsa Svensson, Lina Tirén, Erika Wall

**Affiliations:** 1Department of Health Sciences, Mid Sweden University, 831 25 Östersund, Sweden; asa.m.svensson@miun.se (Å.S.); erika.wall@miun.se (E.W.); 2Östersund Municipality, Children and Education Administration, 831 85 Östersund, Sweden

**Keywords:** adolescent adjustment, health-promoting school, parents, students, school attendance, social-ecological theory, tardiness

## Abstract

Tardiness is a common problem in many schools. It can be understood as an individual risk for future problematic behavior leading to absenteeism, school dropout, exclusion and later health problems. Tardiness can also be examined in relation to a broader social-ecological perspective on health. The aim of this study was to analyze students’, school staff’s and parents’ views on students’ tardiness in two Swedish schools. A focus group interview design was used with 21 school personnel, 21 students in grade nine and two parents. The data were analyzed by using thematic content analysis. The results illustrated the main theme—It depends on…—regarding what will happen if a student arrives late to school lessons. This finding is further explained by the subthemes about teachers’ signals and reactions and the responses from teachers and students. The conclusion showed the importance of organizing the school day more predictably for the students. Late arrival is a sign of shortcomings in a school organization. It is necessary to develop guidelines related to how to handle students’ late arrival based on predictable viewpoints but even more so on how to promote students’ sense of belonging and their interest in and motivation for going to school.

## 1. Introduction

School is one of the most important environments for children and young people. We also know that there is a connection between education and perceived health [1]. Most students report good health, but the number of students reporting mental illness is increasing, as is the number of students who cannot reach their school goals [2]. A systematic review of research on the relationship among health, learning, and mental illness by Gustafsson et al. [1] pointed out several factors related to school performance where school failure and lack of social support were important factors. However, there is less knowledge about the relationship between adolescent adjustments and the role of family involvement in students’ school attendance.

The present study focuses on students’ tardiness within the social and cultural context at school and within families. That is, tardiness can be understood as maladjustment related to various negative effects on academic results and social relations [3,4]. Adolescent adjustment includes attitudes, behaviors, cognitive- and social aspects specifically when it comes to the students’ abilities to adapt within the school environment [5,6]. The students’ adjustment describes the extent to which the student is committed to the school and accepted in the social milieu [5]. Previous research has found positive relations between a high degree of school adjustment and the results in school and social relations within school [3,7].

Regarding adjustment, scholars have pinpointed the importance of the family [8,9,10]. Parenting characterized by the use of reasoning and warmth can contribute as a protective factor when it comes to adolescent adjustment [7,10]. Support from parents is associated with better results in school [8]. In previous research, tardiness has been understood as the expression of risk for future problematic behavior that could lead to absenteeism, school dropout, exclusion, and later health problems [11].

We will examine tardiness in relation to a broader framework based on a social-ecological perspective [12], meaning that the physical, social and cultural dimensions of students’ health evolve in a complex interaction among the environment, organizations, and individuals. The social-ecological theory also appears in the action area of Ottawa’s charter [13], where to build a supportive environment for health is described as actions that “offer people protection from threats to health and enable people to expand their capabilities and develop self-reliance in health. They encompass where people live, their local community, their homes, where they work and play, including peoples’ access to resources for health and opportunities for empowerment”. This action area has been a platform for the development of healthy cities, schools, neighborhoods and other settings for daily life and activities [14,15]. Scholars have found that to be successful as a health-promoting school, collaboration, e.g., with the local community and parents, is crucial [16,17,18]. Hence, we focus not on the specific individual student and his/her behavior but, rather, on the interaction among students, families and school staff from a social-ecological theory perspective related to the school and to the implications useful for developing a supportive environment for students’ health and learning.

### 1.1. Absenteeism

Many factors influence student achievement, directly and indirectly. Students who are absent from school are a major problem in Sweden [19] and in many other countries [20,21]. Swedish headmasters reported that during 2015, 1.7 per mille of all students in grades 1 to 9 were reported absent for a month or more, but 18.5 per mille (or 18,361 students) were reported as being absent now and then (The Swedish Schools Inspectorate, 2016). PISA (Program for International Student Assessment) (2016) reported that nine percent of respondents, aged 15, said they had skipped school one day or more over the last two weeks [22].

The definitions of absenteeism or truancy are not consistent. Kearny [23] suggested the term “school refusal behavior” as an umbrella that covers constructs related to children not going to or coming late to school. He illustrated this as a continuum of truancy going from “school attendance under duress and pleas for non-attendance” to “complete absence from school for an extended period of time” [24], p. 60. The third step along this continuum is tardiness in the morning followed by attendance. However, Kearny meant that a student could move along the continuum and, for example, go from tardiness in the morning to “complete absence…” or to “school attendance under duress…”. Truancy includes all students that have “successfully” missed school and those who tried to miss school but had not yet reached their goal [23]. Truancy must be understood as the complex result of different structural systems around the child, the family, and the school [25]. It begins when the child is young, often in primary school, and it causes harm, especially to the truants themselves; moreover, it is costly (ibid.) Reid [25] also found that parents and students realized the problem caused by the school environment and that teachers believe that parents’ attitudes and the home environment were more influential. In their study, Heyne et al. [26] differentiate between Truancy (TR), School Refusal (SR), School Withdrawal (SW), and School Exclusion (SE). TR captures the problem of a child who does not want to go to school, lacks motivation and has a negative attitude but does not include late arrival. SR differs from TR, whereby the child wants to go to school but finds it too difficult, and this category might include a late arrival, while SW is related to parents versus school-based reasons. However, Heyne and colleagues, differing from Reid’s findings, found that late arrival or missing school is not concealed from the parents [26].

Based on Kearny’s continuum, one should pay attention to tardiness as an imaginary part of a behavior that can develop into school absenteeism, which in turn increases the risk of dropping out [11], school failure, long-lasting associations with crime, and problematic health behavior [27].

### 1.2. Specific on Tardiness

Tardiness is not a new phenomenon, nor is it a problem limited to students at school. Tardiness or lateness to work has been considered a facet of undisciplined behavior related to maturity or motivation, e.g., by Edralin [28]. On the other hand, the use of flexible work time has become more common, and this opportunity for employees appears to reduce lateness behavior [29]. Previous studies on tardiness at school are limited, although it has been a research topic since 1930, e.g., Lockwood [30]. More recent studies, carried out nearly 75 years later by Gottfried [31] in the US, show that tardiness is a risk for school achievement in math and reading not only for individual students but also for their classroom peers. In addition, students who arrive late to the classroom pose a risk for others being tardy (ibid.). In a study of 250 junior high school students, Gottfried [31] found that on average, boys were tardy more frequently than girls, but that half the students in the school had no incidence of tardiness. 

Fifteen-year-old students from OECD member countries were asked about truancy and late arrival at school [22]. The average percentage of students in OECD countries being late once or twice during the last two weeks was 29 percent. The average percentage of students being late more than five times during the last two weeks was 7.3 percent. The average percentage of students who are never late was 55 percent. Never arriving late was at a lower level in Sweden, as reported by 45.5 percent of the students, and 32.6 percent reported late arrival once or twice during the last two weeks. Punctuality was best in Japan and Korea, where between 80 and 90 percent reported that they never arrive late [22], p. 304. Girls and boys are reported in the same group.

In the mid-fifties, D’Amico [32] published an article about teachers’ perspectives on and treatment of tardiness. He found that the way teachers handled tardiness varied: some administered punishments, others took no action. The strategies were sorted into two categories: laissez-faire or punitive. Contemporary studies on tardiness interventions at school are in line with D’Amico’s two main perspectives: (1) reduce negative behavior or (2) strengthen positive behavior. To strengthen a positive behavior, in order to arrive on time at school, Freeman et al. [33] found evidence in their review of skills training and family support but also of incentive-training, where students earned points for both attending class and arriving to class on time. Earned points could be used for movie tickets, school supplies, and restaurants [34]. Other examples include an intervention through which a US high school has a delayed start time, according to Thacher and Onyper [35]. Ugwuegbulam and Naheed Ibrahim [36] relate that punishment is a common strategy to reduce lateness behavior among students in Nigeria. However, they decided to change this behavior by using a fun perspective by introducing a game they played with students who came late to school four times or more within an observation time of two weeks (i.e., ten school days). The game was presented in a welcoming manner: “Good friends, this group is ours, it belongs to us. We are here to play a game in a purely friendly manner. As we play the game, we discuss. Our discussion, which will be on lateness to school issues should be within us, that is, what we discuss here should be confidential” [36], p. 143. The game method has not yet been evaluated, but this is an example of teachers trying to develop new strategies to motivate students to be on time. 

To reduce negative behavior, Tyre et al. [37] implemented the START on Time programme (Safe and Civil Schools Series), including different kinds of punishments in four steps and levels. The first step is sending parents a postcard about a student being late; the second is the student having no lunch if he/she is late again; the third involves activities such as cleaning desks and vacuuming carpets; and the final step, reached if a student is tardy more than 12 times, involves a conference with a parent or guardian, and the student is required to meet with a panel of tribal elders regarding his or her attendance. The evaluation shows that, compared to the average per day and per month prior to implementation, there was a 67 percent decrease in average daily tardiness rates (ibid.).

### 1.3. The Swedish School System

The Swedish school system involves one mandatory preschool year followed by nine years of compulsory school or until the child is 15 years old. The assignment to the schools from the government is to effect compensatory work [38]. This means that schools are responsible for providing equal access to education and for compensating for differences in the students’ capacity. It also means that the entire school system shall minimize the negative effects ofstudents’ backgrounds on achievement and give them an equal opportunity to learn [39,40]. The Swedish school organization is responsible for creating a good learning environment for students’ personal development and the development of their knowledge [38]. This includes promoting efforts and projects to increase students’ attendance and thereby minimize absence in education. It requires systematic work based on a regular review of their own organization and content based on knowledge of the factors that promote student learning and development. The efforts must include all students.

How the school’s working environment is designed can in itself promote presence or contribute to absence. A good learning environment, with teaching adapted to the individual student’s needs, promotes motivation to participate in education. The participation of students in the work of promoting presence can be effected in many different ways. They must participate in the work of creating a common approach to late arrival and determine what will be considered an invalid absence [38,39].

In summary, we have discovered that there is a gap in research in regard to studies about tardiness in school, even though it is well known that early identification of student tardiness increases the opportunities of detecting and preventing a problematic behavior that could eventually lead to absenteeism, school dropout, exclusion, and later health problems [3]. In addition, there is a lack of research on how absence and/or tardiness among students effects education and the work environment for the other students and the teachers, and vice versa [41]. For example, tardiness is often explained in relation to individual behavior, such as children’s sleeping behavior [42], and is seldom related to a school context. We argue that tardiness is overlooked in research and practical work related to health promotion in school. Moreover, we claim that tardiness, as a phenomenon, is overlooked. In terms of a social-ecological perspective on health [43] and a critical perspective on health promotion [44,45], it is crucial to include perspectives on who is most affected: the student. However, most research in this field appears to be based on statistical analyses of absenteeism [19,46].

Thus, there is a research gap in regard to including the student’s own perspective on attendance, absence and tardiness, as well as a lack of studies that include stories of those who are part of the social setting of students at risk of absenteeism (i.e., teachers, school administration, personnel working in health-related capacities among students, parents, etc.). Given this background, the goal of the present study is to contribute knowledge about tardiness that can be useful in strengthening practical work at school from a health promotion perspective. Its aim is to analyze students’, school staff’s and parents’ views on students’ tardiness.

## 2. Materials and Methods

To study students’, school staff’s and parents’ views on students’ tardiness, a focus group interview design was chosen. To this end, we created groups of participants and provided instructions regarding discussion themes [47,48,49]. Examples of interview questions to parents and teachers include: “What do you think promotes students’ school attendance? and “How do you perceive the responsibility of the school in relation to students’ school attendance?” To the students: “Does it ever happen that you arrive late to school? ”and “When does it feel good to come to school?” Two researchers participated in each interview: one led the discussion while the other handled the audio recorder, took notes and observed [50].

### 2.1. Participants and Procedures

This study took place from January to June 2017 in a mid-sized town in the northern part of Sweden. The schools were located in a suburban area primarily comprising middle-income households. The participating schools were both compulsory schools. From the age of six, Swedish children are allowed to start primary school and most of them do so; from seven to fifteen years of age, school is compulsory. The first school (A), with approximately 200 pupils, went from primary class to grade 5. The second school (B), with approximately 350 pupils, included grades 6 to 9. The headmaster at the school with the younger children (A) selected nine school staff members (teachers, assistants; two men and seven women) who participated in a focus group lasting one hour.

At the school for students in grades 6 to 9 (B), the research group introduced themselves to the staff, presented the study, and asked for volunteers for the focus groups. Staff were also given an information letter and a list was left in the staff room so they could sign up for the focus group. A total of twelve school staff members accepted the invitation. They were divided into two groups: one with the school health team (SHT, including the headmaster, school nurse, psychologist, school welfare officer, and special pedagogue; two men and five women) and one with teachers (two men and three women). All parents at school B were invited by e-mail, and three of them accepted. One of them was prevented from participating, so the interview was carried out with two parents.

Based on Swedish ethical law, teenagers aged 15 are mature enough to decide on being research participants [51]. Therefore, only students of that age were invited to participate in the study. This was also decided because students in grade nine have the most years of school experience. One of the authors introduced the study to the students in class and asked for participants. The author also spoke with the students about the study during breaks and reminded them of the study and invitation. Twenty-one students in grade nine (14 boys and seven girls) accepted the invitation. Mindful of Wernersson and Ve’s [52] finding that boys tend to dominate girls in groups and of the overrepresentation of boys who wanted to participate, we generated two single-sex groups and one mixed group. The students decided on their own if they wanted to be in the single-sex or mixed group. Group 1 included nine boys, group 2 included five girls and group 3 was a mixed group (two girls and five boys).

All interviews except one were held during the day in a separate room in the school building. The parents were interviewed in the evening in the school library. The two focus groups with teachers lasted from 60 to 90 min, the focus group with the school health team lasted 60 min, and the focus group with the parents and pupils lasted 30 to 40 min. All interviews were audio-recorded and transcribed verbatim by an independent transcriber.

### 2.2. Analysis

The material was analyzed in order to determine which themes best describe the content, by using a process inspired by Braun and Clarke [53]. In the analysis, all transcripts were initially read by three of the authors independently. Based on these readings, themes describing underlying aspects of the stories told were constructed. That is, specific parts of the transcripts were highlighted and grouped as the analysis was conducted. During this process, delimitations were constantly discussed and challenged in the research group. The researchers also noted down their thoughts and interpretations while reading the material. The themes were carefully reviewed several times before arriving at a result. One overarching theme and three subthemes are the focus of the reported results. These subthemes are not mutually exclusive; rather, the subthemes overlap, as one leads to the other. Together they tell a story about students’, staff’s and parents’ views on tardiness. The themes should be viewed as representing patterns of shared meaning throughout the interview transcripts. The original spoken language in the interviews was Swedish. At the end of the analysis process, the results, including quotations, were translated into English. 

### 2.3. Research Ethics

At the outset, all participants were given oral and written information about the project and were informed of their right to withdraw at any time without explanation or negative consequences. All participants signed an informed consent.

The Regional Ethical Review Board in Umeå, Sweden, found that the study was not covered by the Ethical Review Act and did not consider the study as an issue of ethical concern (2017/154-31).

## 3. Results

We began our study with the concept of being present in or attending school and class. During an interview with two parents (school B), both mothers, the importance of students being on time for school and class were emphasized. The parents viewed school attendance from a broad perspective and on various levels. That is, punctuality was understood as a skill for future life and a way to be respectful to others; however, they also found that students must bring books, bags and other items with them, in addition to being on time for their lessons:


*Yes, that they are in place, at the lessons. And on breaks and lunches too, of course. So, they are in place on school days, that’s what I think.*
(Mother 1)


*Yes, that they are on time and are here (in school) when they should be.*
(Mother 2)


*Yes and have what they need to bring to school, like books, clothes for physical education, bags, or whatever it is.*
(Mother 1)

The parents’ statements on students arriving on time, collected at the beginning of the study, inspired us to further explore how tardiness was viewed by students, parents, and school staff. The analysis relates to how tardiness is viewed in the material. The thematic content analysis showed that perspectives on arriving on time at school are framed by the main theme “It depends on…” This main theme is supported by the following underlying subthemes: (1) Signals, (2) Reactions, and (3) Responses. Here, the main theme will be presented first, followed by the subthemes. 

It depends on… 

Throughout the material, ”It depends on…” permeates the analysis. Students’ actions were viewed as depending on who the teacher was and how he/she behaved when they arrived late. Moreover, students’ actions were influenced by many factors: the importance of the school subject, the motivation towards school in general, support from home and time of the day. Similarly, the teachers’ understanding and handling of late arrival was related to a number of various circumstances: the teacher him/herself, the student, the circumstances of the late arrival, and lateness frequency. When it came to the parents’ views on the issue, the answer was it depends on…; thus, they told their stories about families and teachers. However, the staff on the student health team (SHT) at school B, did not explicitly mention tardiness but discussed school attendance and truancy in general terms and mentioned the importance of collaborating with parents. That is, from their perspective, it depends on the parents if students arrive at school on time or not. Parents were seen as either supportive or insufficient in their role.

In relation to the main theme, and shedding light on the complexity of tardiness, the developed subthemes in the analyzed material is as follows: tardiness depends on what signals are perceived regarding being on time; how tardiness is met by reactions of various kinds; and how students (should) respond to expectations and actions related to tardiness. Notably, these subthemes should be seen as a product of the analysis aiming to describe the material in the best way possible to answer the aim of the study. The subthemes should, therefore, not be understood as mutually exclusive but, rather, as mirroring the complexity of the issue.

### 3.1. Signals

The first subtheme, signals, includes accounts of unclear signals about the importance of being on time. The signals relate to expectations and norms and include issues of structure, predictability, and belongingness. 

The mothers saw no consensus among the school staff regarding how tardiness is viewed and handled in school. They explained the various ways in which teachers dealt with students as disparities in teachers’ tolerance and as normalizing problematic behavior. The mothers requested higher expectations from the school regarding students being on time. They expressed that it is important from a security aspect that the school notice students arriving late. As a parent, you want to know that your child is at school and nowhere else. Furthermore, they believed that the students themselves would like to feel they were expected to be on time and that this emphasis is about respect and routines.


*A time is usually to be held to for a reason. So if you do not show up or do not bother being on time, then you do not show respect for it. Everything I think is about consideration, respect, and cooperation. We are numerous at the school; it should work for everyone. And then slipping in when you feel like it or not showing up at all, this affects oneself first and foremost but affects others as well. You get into it, there just has to be a routine...In our family, we are useless at being on time, but…you have to try. And then there are surely families who, ‘but my goodness, come on time, it is well a worldly thing, there are worse problems in life’. But somehow, the school must uphold routines. If you don’t have a routine in school...and do not have one at home, where do you have it, if you are a youth? You have these two platforms and then perhaps a hobby. And then there must be security in this.*
(Mother 1)

The mother’s story illustrates the importance of respect for others through punctuality but also of the school itself as being responsible for maintaining structure, which can be understood as helping families struggling with punctuality.

The teachers in the two focus groups, at school A and B, had different views on late arrival. Some of the differences were between the two schools or related to what grades they taught. Some teachers expressed that they expected students to be on time, while others did not seem to care if students were a few minutes late. It seemed to depend more on the situation than on the student. The teachers were said to be more understanding if their class followed, e.g., physical education, as well as if the student was usually on time. If a group of students arrives late, one teacher said, she directs her attention instead to those who arrived on time and praises them. Furthermore, in the material, arriving late seemed not to be viewed as being equivalent to being absent from an entire class.

The teachers said that being on time did not seem to be important for some students. This idea was extended to families; that is, teachers in the lower grades (school A) said that not all parents found it important to help their children be on time. These teachers also expressed more responsibility and caretaking that fosters a sense of belongingness. One of the teachers said, 

*So that you...And that you get in contact directly, so...*(to the Guardian), *if a student does not come to school, for example, and you have not received an illness report or any other absences, then you must call immediately, of course.*

The school staff in this school also discussed the importance of being clear with the students about what is expected of them and what happens when they do not live up to those expectations.


*They begin arriving a little bit late from breaks, linger, don’t take things seriously, or the student left home a little bit late. And of course, it affects others too, those who are always there on time [laughter]. So, of course, that is part of the reason why we have tried to work on it.*


The students’ stories (school B) also include descriptions of blurry signals regarding expectations about arriving on time. Depending on the reactions from the school, the students maintain there are no clear norms regarding punctuality or being on time for lessons.

### 3.2. Reactions

The second subtheme is about how tardiness is handled by teachers. In the focus group interviews with students, the most common description of what happens when they are late was that no one reacts. In one of the focus groups, the girls were asked what would happen if they arrived late. The reply was that the teachers most often do not respond at all to late arrivals; however, it depends. One of the girls said, “It depends. Sometimes they do not react at all”. Another girl continued, “Often” and a third girl said, “I would say that they do not react at all.” Similarly, the boys said “It depends”, “It is different (if and how they react)”.

That is, the students pointed out that sometimes being late was met with no reaction but, if handled, the reactions vary from teacher to teacher. Some simply noticed it and said “Welcome”, while others punished latecomers by screaming and shouting at them. Therefore, teachers’ reactions to late arrivals were seen as being unpredictable in some ways. The material analyzed showed that these variations are understood by the students as relating to a specific teacher (or other personnel) or, on the other hand, to a specific student. That is, all students are not treated equally. These variations recurred independently of interviewees (boys, girls, parents; notably, the SHT did not speak about late arrivals at all).

Students expressed that late arrival was sometimes recorded in the electronic system but sometimes not. If the students did not notice such a record, they interpreted it as “no reaction” from the teacher. Registration in the electronic system decided whether the student was judged as being late or not; this was in some ways predictable for the students, e.g., which teacher would register tardiness, but even among those noting it, the handling of the situation varied. Some teachers at school B noted every single tardiness minute while others recorded only if someone was really late, although what was considered “really late” differed among teachers. The teachers at both schools and the SHT spoke about a 20-percent limit to school absence measured from the beginning of the term. When a student exceeded that limit, the SHT initiated a meeting with the parents. 

Some teachers (B) said they used a strategy to ignore students when they were late to class: they just went on with their lesson without paying attention to the latecomer. Further, some teachers simply asked the latecomer, “why so”. The analysis shows that what the students define as ‘no response’ to a late arrival sometimes means a response but a non-social one. Students relate responses to social reactions in the relationship between teacher and student. One of the boys did not want to be asked for an explanation as to why he was late: “It is good if they ask why you are late, so you can explain… There may be reasons”. A girl wanted more caring questions from the teachers: “They don’t ask us ‘Yes, why are you late?’ or ask for the reason why we’re late. They just note the absence. They are so used to it”. The girls had different opinions about the teacher’s caring questions; although they do not want to be absent for a long time without the teacher noticing, they wanted to be missed. In addition, in the mixed group, questions about being late were considered preferable.

Although a lack of reaction to tardiness was found in the material, the opposite was also interpreted from the stories told, with the analysis showing that powerful reactions from teachers are also important. Teachers who felt provoked or interrupted sometimes reacted aggressively. Their various stories about situations involving late arrival showed how teachers feel about youths. Some teachers (school B) described students “flinging” and “kicking on the door”, or simply waiting until the lesson began and then coming in. Other teachers, at the same school, based the stories of their ambition to create a welcoming classroom with opened doors and on thoughts about why students arrive late or drop out of class or even from school.

In regard to reactions to tardiness, doors played a central role in the material from school B. The students saw a closed door as a clear signal that the lesson had begun; on the other hand, the door could be open even if the lesson was in progress and the students could just step inside. This was experienced by the students as being unclear and showed that lessons could begin in various ways depending on specific teachers and how they set up their own lessons. The girls understood that teachers do not want to be interrupted when they begin a lesson:

Respondent girl:
*She held a review… then it is another thing. If she is holding a review.*


Interviewer:
*If she has a review, then she reacts?*


R girl:
*Yes, then she doesn’t want people knocking on the door: it disturbs everyone else and also her review.*


Other teachers not only close their door but they lock it, which extends the length of the absence that will be reported: 


*Some teachers, they do this... If you are a little late, you cannot get in; then, you get marked even more absent than you really were… So, you really were earlier than the report shows because the teacher didn’t let you in. And then you want to be on time. You do not want to get an unnecessarily long absence. The teachers lock the door, not just close it.*


Furthermore, in one focus group with teachers (school B), locking doors was pointed out as being a controversial activity related to security aspects. “The rules about locking doors have been changed”, one teacher said, after a crime was committed in a Swedish school two years earlier when a young man killed a student and a teacher and seriously hurt a group of young students. The other teachers in the group took no notice of this comment and continued describing their strategies in handling late arrivals and how they felt provoked by students knocking or kicking on the door to get in.

Unpredictably, students mentioned the strong reactions from teachers. In some cases, they said that the teachers reacted more strongly to someone arriving late and knocking on the door. It seemed to depend on what was going on in the classroom at the time. The students said that teachers seldom reacted if the students in the class were working individually while someone arrived late. On the other hand, they pointed out that some teachers not only respond negatively to being interrupted but also to other events in the classroom. There were accounts of teachers frightening students by their behavior, one student reporting, “One may be scared of some teachers, so then one doesn’t want to be late. They may be…very angry”.

The students who encountered screaming and shouting teachers said that they could avoid such treatment by being on time, but their narrations also contained accounts of various behaviors among teachers, behaviors that are not always predictable. However, in one of the groups, when ‘shouting’ teachers were mentioned, this was not considered a constructive way to handle latecomers, even though students found they were seldom late to classes conducted by those teachers: 

R boy:
*It is good that they record an absence but it feels excessive that they stand there and scream.*


R boy:
*Yes, exactly. It is our choice anyway if we arrive late or not. Then, he can record us as absent, nothing else. It affects us. Not like this: stand and mess just because you are late.*


I:
*What do you others say about this?*


R:
*Same here. Just go in and sit down, and then they record you absent, and so…yes, it will be like that. Then, there is nothing more. You shouldn’t have to talk about why you are late and so on; it is just unnecessary.*


I:
*At the same time, you said that the teachers who get angry, those are the lessons you go to to avoid it (the teacher’s negative behavior).*


R boy:
*Yes, then it works. Thus, the students will be on time.*


R boy:
*Yes, I was thinking the same thing.*


I:
*And then, the teacher has achieved his/her goal?*


R boy:
*Yes (from many).*


From the above quotations, it is clear that the students believe that late arrival is simply their own choice and that they do not see themselves in relation to others. However, some of the students could see the contradiction in wanting the teachers not to react but at the same time coming to class to avoid an aggressive teacher.

### 3.3. Responses

The last subtheme focuses on responses; that is, issues that strengthen the willingness to be on time and that contribute to tardiness and absenteeism. In the interviews with the students, various perspectives on the relevance of being on time were emphasized. Some expressed the view that classes important for high school were prioritized, others prioritized classes with teachers they found kind and interested, and still, others mentioned the importance of being with friends. Students also found that the short time was a motivation for going to school and being on time.

For the most part, students wanted to avoid such aspects that create discomfort. They described angry and shouting teachers and angry parents, but there were also aspects that could be a motivation in the long or short term. One boy, who had been truant and tardy a lot, said: “I had to start thinking about getting into high school. So, school started to feel more important”. He was motivated by thoughts of his future life.

Some students discussed being late as their own choice, according to how seriously they view school and education. They were more motivated to attend some of the lessons. That is, they were found prioritizing various subjects; some were noted as being more important, and others were considered more boring. Thus, students said they were seldom late to those prioritized classes and more often arrived late to other classes.

There were also differences in maturity among what was considered motivating aspects. One boy was motivated by more urgent needs satisfaction: to avoid conflicts with his mother, he had coffee at school in the morning; nothing else motivated him:

R boy:
*Yeah, I don’t know. Not so damn much. I have to come here; otherwise my mom will not be so damn happy if I’m home every day.*


I:
*It is your mom….*


R boy:
*Yes, she forces me to go to school.*


I:
*Yes?*


R boy:
*Yes.*


I:
*So, it’s just about force, never any strong desire or anything (to go to school)?*


R boy:
*No, damn it. You get coffee here.*


I:
*You get coffee? You don’t get that at home?*


R boy:
*Yes, damn it, but I have to make it myself; I can’t stand it.*


In order to understand what motivated students to make them be on time for school, or even go to school, we asked them to describe their “dream school”. They told stories about starting later in the morning, better teachers, better technical equipment, better food, and more variety and flexibility.

The relevance of motivation was found in the parents’ stories as well. However, parents’ perspectives on motivation were related to the future of their own child. That is, school is important in obtaining a higher education and/or a job. In the material based on the focus group interviews with school staff in both schools, this perspective was not made explicit. When the staff (schools A and B) talked about motivation, their stories were mostly related to how they did not find that students and/or parents prioritize school. The staff questioned the students and/or parents when it came to prioritizing being in school and/or on time for class.

When the school staff (A) were asked to describe an “attending” student, they mentioned individual aspects such as being curious, motivated, healthy, idea-rich and willing to learn. In addition, aspects related to their social environment were mentioned: being pushed by their parents, living in a family engaged with the school, other activities outside school and having friends. The teachers in the secondary school (B), on the other hand, noted the importance of being welcoming and showing that they, as teachers, were happy to see the students, and of developing good relations between teachers and students. In terms of pedagogy, they tried to make education “old school”, that is, predictable and without too much involvement of the students, in order to make it easier for them. From their perspective, this was a way to motivate their students.

## 4. Discussion

The results showed that tardiness, viewed by students, parents, teachers and other groups working in the school, including the student health team, is a complex issue. However, based on various *signals* relating to how important attendance was understood to be, how eventually tardiness was reacted to, and the students’ responses, the pattern we found illustrates the unpredictability encountered by the students, whereas tardiness itself was not viewed in similar ways by the focus groups, nor was it handled equally. These results are discussed below, first in relation to the theme developed from the analysis and then based on our main theme and how it could be understood from a social-ecological perspective.

### 4.1. Signals

Parents and students interpreted the signals from school as indicating that punctuality was not that important. This can be understood as meaning that teachers have no expectations of you (as a student) to be at school on time. Previous studies show that school attendance is a sign that the school is well structured and predictable for the students [26]. Buhler, Karlsson, and Österholm [54] found it important to promote school attendance by starting and ending the school day in the same way, every day. Teachers ignoring latecomers also indirectly send out signals that you as a person are not important. This result is in line with previous findings [55]. Expectations of how people act in different situations produce norms related to how we behave in a specific environment and indicate what the culture is. Signals can be used to create a sense of belonging or create a feeling that you do not belong to this place or culture. Belongingness has been found as essential in the health-promoting school model described by Rowe and Stewart [56].

To feel included in and connected to school is important for health and school achievement [57]. Wenzel [58] found that perceived support from teachers measured as, for example, “My teachers really care about me” and “My teachers like to help me learn”, was a positive predictor for student motivation, school interest, and class interest one year later.

### 4.2. Reactions

Latecomers encountered different reactions. Schools have no shared and clear policy or rules to handle late arrival: some teachers locked the door, others registered the tardiness in the electronic system, some shouted and still others took no notice of the latecomers. The predictability became even more unclear from the students’ viewpoints, because sometimes the same teacher reacted differently to the same situation involving different students. This result shows that not too much seems to have happened since D’Amico [32] carried out his research and found no consensus among teachers. On the other hand, treating students unequally might be how teachers interpret compensatory assignments [39]. Swedish compensatory assignments offer support related to the student’s needs for equal access to education. With students coming from a vulnerable family and/or environment or with at-risk students, teachers probably want to be welcoming and caring in order to mitigate their vulnerability. The results indicate that teachers from school A saw the students’ wellbeing and school achievement as a part of their family and the entire living situation.

Teachers who sometimes emphasized the differences among students created an unjust school environment. This was also found in a Swedish [59] qualitative study, where the students reported feeling like “black sheep” or failures, while other students were described as being teachers’ pets or high achievers. The “black sheep” described how teachers’ behavior, for example, shouting and screaming at them, had affected their self-esteem and interest in schoolwork. In addition, Banfield, et al. [60] described “offensive teachers” who humiliated students, picked favorites, and were rude or sarcastic, which in turn affected the students’ wellbeing. Handling tardiness by being offensive to students is in line with D’Amico’s [32] description of reducing negative behavior. On the other hand, Freeman and colleagues [33] found evidence that skills training and family support, in addition to incentive-training, to increase the level of student punctuality strengthened positive behavior.

The students in our study asked for more caring questions from the teachers and for greater openness to being exposed to unforeseen events. This could be seen as young people’s expectation of a relational perspective on teaching [61]; in addition, a relationship with or interaction between teachers and parents was also mentioned by mothers, an idea related to the fact that the primary means of communication was expected to be via the electronic system.

Mothers, students, and teachers alike mentioned security aspects but related to different situations. Students and mothers wanted someone to start looking for latecomers because something may have happened to them on their way to school. Teachers in the lower classes also thought along these lines. Secondary school teachers, on the other hand, mentioned security in relation to school attacks and the need to keep the students in a safe place, away from external threats. School shootings or school attacks have been a reality for many years in Western countries [62]. The risk that something would happen to a student on her or his way to school is more likely than a major attack in the school itself. The responsibility of parents and of the school towards the students is an ongoing topic of discussion in Sweden, but in some ways, the rule is clear: parents are responsible for their children arriving at school on time. The school is responsible when the child is at school, but the responsibility is twofold between school and parents [63]. This could explain the differences in the focus of the teachers, parents, and students.

### 4.3. Responses

Predictability, a culture that creates a sense of belonging by giving caring and supportive signals in school, affects whether the students arrive on time or not. Findings [64] reveal that a negative sense of school belonging has a negative impact on intrinsic motivation and perceived learning.

The students (mostly the boys) saw themselves as solely responsible for arriving on time, and this was only related to their own decision. On the other hand, their responses showed that their decisions were influenced by teachers’ behavior, parents’ expectations, future plans for higher education and, in some cases, simply by the short time period at their disposal. Another finding was the school staff’s discussion about motivation. Motivation was described from an individual perspective as a characteristic of the student or coming from a supportive family. None of the teachers mentioned motivation as the responsibility of the school organization or as part of pedagogy, even though a number of previous studies have noted that motivation is part of classroom pedagogy [65]. The teacher-learner relationship is also an important factor in creating engagement and promoting educational outcomes [66].

We interpret this finding to mean that this school had not created a sense of belonging for the students. Nor had they, from the students’ viewpoint, created a desire for learning. Both students and parents asked for better interaction and communication with the teachers and the school. The increased digitalization of school involves the risk of losing communication and interaction with parents if all primary communication is effected through texting and through registration in an electronic program. Parental engagement in school has been shown to be important for school achievement [67] and for a health-promoting school [16,17,18].

The results show that teachers do not differentiate between school refusal behavior and truancy or school withdrawal, and neither does the school health team. They focus more on absenteeism percentages and inputting these into the electronic system to begin analyzing the problem if the absenteeism level exceeds 20 percent. The reason for tardiness must form part of the analysis that determines effort. Universal and individual efforts do not contradict each other: both are needed. In this article, we focus on how the school can be a supportive environment for students and their families in regard to students arriving on time. To promote school attendance, specifically punctuality, one must examine the student and his or her individual behavior within a broader social and cultural context in which the student, the family, and the school interact. The main theme, “It depends on…”, will now be discussed from a social-ecological perspective.

### 4.4. It Depends on…

According to social-ecological theory [43], different environments influence our behavior and health. People’s thoughts, behavior and perspectives are transmitted among these environments. We move between different environments: our family, neighborhood, school, work and so on. From a broader perspective, the child’s (and school staffs’) home and neighborhood is also part of the system whose components interact with each other [43,68]. Earlier studies have shown that young people see the interaction between the school environment and their home environment as important for their health [59]. This is in line with the idea behind health-promoting schools [69] and the entire school approach [70].

The main theme “It depends on…”was developed to describe the various aspects influencing how teachers give signals and react in relation to tardiness and how this reaction elicits responses among students. The interpretation of “It depends on…” provides a picture of an unpredictable situation for both students and parents. As mentioned above, predictability and structure during the school day have been shown to be important factors for school attendance, especially for children with school-refusal behavior [71]. In addition, other studies illustrate the importance for all children of consequent, clear and equal rules and norms [72]. The mothers interviewed in the present study stressed the importance of the school as being a place for structure and safety and encouraging an “arriving on time” culture, based on the fact that some children live in a vulnerable situation in families with many social problems. This finding is in line with the directive from the Swedish government about compensatory assignments [39,40].

The social-ecological theory explains how specific cultures develop in different environments. The students move between school, their homes, and places for spare-time activities. These places are culture carriers, but the students affect and are affected by these different contexts when they move and interact within them. The common setting for the students is school, while the other settings differ more or less, given that homes, neighborhoods and to some extent, settings for sports and other activities, differ according to the families’ socioeconomic position [73]. Schools in Sweden are obliged to provide equal access to education and shelter, to be compensatory and to strive to equalize inequalities, although sometimes the opposite is the case, even in Sweden [74]. In line with this, the teachers of younger students see their work in encouraging punctuality as a part of their educational assignment, as an interaction between how they organize their schoolwork and how the students respond, and how this, in turn, affects the classroom environment. This example illustrates the ecological system and the constant movement within that system.

Individualized education, on the other hand, where teachers say, “Take your books and continue where you are”, creates a view of the individual student as an isolated island, without any connection to the environment. This reductionism assumes that a system can be broken down into single components, which is the opposite of a system in which parts interact and must be understood in relation to the whole [75]. However, why should students arrive on time if they do not assume they relate to others or to the environment?

Lastly, the results from our study indicate that tardiness is understood by the participants as a problem related to both the school and the family, also observed in research on adolescent adjustment; illuminating not only the students’ ability to adapt with school environment [5,6] but also the impact of parenting on these abilities. That is, positively engaged parents can contribute to adjustment [7,8,9,10]. That means, vulnerable families need more support from schools and maybe also social welfare services, on how to become more involved in their children’s education [76].

Altogether, the social-ecological theory [43], as well as the school adjustment perspective [5,6,8], illuminates the importance of family involvement. To promote school attendance among students, a social perspective including the whole social context of the adolescent is necessary.

### 4.5. Methodological Discussion

A focus group interview design was chosen for the present study. Students, school staff, and parents were invited to participate in order to provide different perspectives on the topic of our study. School staff were deliberately interviewed in groups based on their profession (i.e., the school health team were interviewed separately from the teachers) since we assumed that they would have different views on tardiness depending on their professional role and experience with students. It turned out that they did speak about tardiness in different ways.

Two parents agreed to be interviewed. Preferably, one to two focus groups (about 5 to 10 parents) should have been performed to give equal weight to the stories of students, school staff, and parents. This approach would have strengthened the results of the parents’ perspectives on tardiness in the present study. The two participating parents provided valuable insights into their views on tardiness. In fact, it was the interview with the parents that introduced tardiness as a topic of interest and led us to study it in more depth.

Recruiting students ensures that adolescents from various backgrounds were included since school is mandatory. However, students absent from school when recruitment was carried out may not have received the information and were thereby excluded from the focus groups. Students absent at the time of recruitment are probably more likely to be absent from school in general and, if interviewed, would perhaps provide important insights into the reasons behind tardiness. Conducting individual interviews instead of focus groups would perhaps have decreased the risk of peer pressure among the students when interviewed and made it more likely that they spoke their minds. On the other hand, focus groups with peers could be a more comfortable venue for students than being alone with the researchers. In our opinion, interviewing the students in same-sex or mixed groups based on their own choice made it more likely that the participants were comfortable and could express their opinions.

In addition, focus group interviews are preferable when seeking insights into meaningful themes embedded in discussions of the topic chosen. The included schools, school staff, parents and students were not representative, although we included both a primary and a secondary school, and the staff members interviewed represented all key professions.

Our study found that a thematic analysis provided the best opportunity to answer our research question. Criticism of a thematic analysis includes individual accounts and language use being lost, and the flexibility of the method making it difficult to know what aspects to focus on. We aimed to present patterns of shared meanings. It turned out to be difficult to create themes that were mutually exclusive, as recommended by, e.g., Braun and Clarke [53]. Nonetheless, we believe the thematic analysis is the best analytical categorization for the material in our study. The present study used social-ecological theory and health-promoting perspectives as a framework to strengthen our interpretations. The involvement of three researchers in the analysis process and a continuous discussion during all steps of the analysis strengthened the process and the credibility of our results. All authors agreed on the final themes. We have also described our methodology and analysis in detail to allow readers to form an independent assessment of credibility. To judge our study’s resonance, the preliminary results were presented to and discussed by some teachers and the principal from another school for feedback. The results made sense to them.

## 5. Conclusions

The picture that emerges from our analysis shows both differences and similarities among the groups participating in the study. Parents and students shared the same idea about how schools handled lateness, but their views on responsibility differed. Some of the students saw only their own responsibility as a factor affecting punctuality. The inability to see their role as part of a greater whole was more prevalent in secondary school teachers than in teachers in the lower grades. Late arrival is a sign of adolescents’ maladjustment from a holistic viewpoint, in a school as an organization created to interact with students, families, and school staff.

The results of this study imply the importance of organizing the school day more predictably and with a better structure for the students. Such predictability can be expressed by starting the school day at the same time and in the same manner every single day. Further, the staff needs to agree on a common policy on handling tardiness. The result also indicates that schools must further develop their effectiveness in relation to both students and parents. That is, they must strengthen their work in promoting students’ sense of belonging and in relation to students’ interests and motivation to come to school. In dealing with tardiness especially, it is important to interpret this behavior as a signal understood in a greater context.

In further research, we suggest developing a model of how to handle student tardiness based on predictable standpoints. This model should be developed and tested in collaboration with researchers, teachers, students, and parents.

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
