# Peer review of "On Time: A Qualitative Study of Swedish Students’, Parents’ and Teachers’ Views on School Attendance, with a Focus on Tardiness"

_ijerph, 2020, doi:10.3390/ijerph17041430_

Round 1

Reviewer 1 Report

Thank you for the opportunity to review the paper:  On time: A qualitative study of Swedish’ students’, parents’ and teachers’ views on school attendance, with a focus on tardiness.

The present study aims to analyse students’, school staffs’ and parents’ views on students’ tardiness in two Swedish schools. A focus group interview design was used with 21 school personnel, 21 students in grade nine and two parents.

This is an interesting manuscript that meets the quality requirements demanded by the journal.  So, it has several strengths, such as the clarity with which it is written and the use of sophisticated theory and statistical measures. However, the study is still flawed.

As an opening point, the title clearly reflects the theme of the manuscript, being in line with the guidelines for authors. The extension of the title is adequate, and the purpose of the study is sufficiently clear. The summary gathers the key points of the research and adjusts correctly to an abbreviated and complete presentation of the context, responding to the purpose of the study. For clarity reasons, perhaps research questions should be enumerated and posed as questions. The expectations/hypotheses could follow right after each question. The method and material used is adequate and well described in the work. The size of the sample is representative and is adequate to carry out the statistical tools proposed. In the discussion the correct revision of the most relevant results is carried out, as well as the comparative analysis with respect to the contributions of other authors. The conclusions are supported by the data presented in the results, and the inclusion of the final section. However, no relevant educational implications are provided in this study. Therefore, it would be appreciated if the authors deepen them. Another important issue is the references. Reviewers are kindly requested to review the references

Obviously, this manuscript has some flaws that may render it unsuitable for publication. I encourage the authors to consider these comments and those of other reviewers to decide on next steps based on available time and information.

I hope my comments may be of help to authors in their work.

Author Response

Dear reviewer,

 First, we like to thank you for taking your time to read and make comments on and suggestions to improve our manuscript.

We have put your suggestion as comments in the matrix below and answered how we have responded to them.

Point Comments Responses
1. "For clarity reasons, perhaps research questions should be enumerated and posed as questions. The expectations/hypotheses could follow right after each question" We decided not to use specific research questions and hypotheses, in line with a qualitative research paradigm. The qualitative study is exploratory by its nature, and the aim is therefore often more vaguely worded compared with the aim of a quantitative study.
2.  ...no relevant educational implications are provided in this study. Therefore, it would be appreciated if the authors deepen them.

On suggestions from the reviewer we have included educational implications in the section with the heading Conclusions and from that rewritten this paragraph. See line 637-657.

3.  Another important issue is the references. Reviewers are kindly requested to review the references.

We have made some corrections but let the editor judge if they are written in line with the journals house style and give us some advice if they something more needs to be changed.

Reviewer 2 Report

I recommend to publish this article. It opens an interesting topic to discuss and compare in different countries, tardiness. And I think the method is coherent with the social ecological approach.

Some suggestions are the following ones: 

Results could be improved if the quotes of the participants are differentiated from the comments of the authors. This is clear in lines 279, 304, 371. 

There are phrases which should be reviewed lingüistically in 344, 378 and 459.

I don't think section 4.5 add anything relevant. I would remove it. 

I would jusfify better in the results some relevant theoretical topics that are presented in the discussion, such as the importance of a sense of belongingness, the difference in the educational conceptions of the teachers... including a relational stance or not, including a shared responsibility or not to create motivation, etc... I would also suggest to differentiate better School A and B in the results to prepare the discussion.

I really consider an article worth reading. 

Author Response

Dear reviewer,

First, we like to thank you for taking your time to read and make comments on and suggestions to improve our manuscript.

We have put your suggestion as comments in the matrix below and answered how we have responded to them.

Point Comments from the reviewer Responses
1. Results could be improved if the quotes of the participants are differentiated from the comments of the authors. This is clear in lines 279, 304, 371 We have made quotation marks around the quotations, and made the sentences with italics to clarify where they starts and begins. However, the journals house style on how quotes are presented in a qualitative study could be with block quotations for longer quotes. If the editor suggests this, we will change them. 
2. There are phrases which should be reviewed linguistically in 344, 378 and 459. We appreciate the reviewer's careful reading and regret that this happened despite the use of a professional editing company. We have now change the sentences linguistically.
3 I don't think section 4.5 add anything relevant. I would remove it

In qualitative research, the method discussion is important to proof trustworthiness and let the reader examine, but if the editor agree with the reviewer, it can be excluded.

4. would jusfify better in the results some relevant theoretical topics that are presented in the discussion, such as the importance of a sense of belongingness, the difference in the educational conceptions of the teachers... including a relational stance or not, including a shared responsibility or not to create motivation, etc...

Our analysis is no comparative our aim was to analyze the view of a students’, school staffs’ and parents’ on students’ tardiness.

We have now clarified line 269-270 that belongingness is a part of signals related to expectations and norms and also to care taking line 302. We have also added a sentence in the discussion, citing Rowe and Stewart 480- 481.

5.

I would also suggest to differentiate better School A and B in the results to prepare the discussion.

We have marked citations from school A and B and hope this will help to differentiate the schools.